systems biology/behaviour/health and disease and epidemiology

anthrax, ungulates, bacterial diseases, *Bacillus anthracis*, locally infectious zones

**Author for correspondence:**
Jason K. Blackburn
e-mail: jkblackburn@ufl.edu

# Ungulate use of locally infectious zones in a re-emerging anthrax risk area

Morgan A. Walker[1,2], Maria Uribasterra[1,2], Valpa Asher[5], José Miguel Ponciano[3], Wayne M. Getz[6,7], Sadie J. Ryan[2,4] and Jason K. Blackburn[1,2]

[1]Spatial Epidemiology and Ecology Research Laboratory, [2]Emerging Pathogens Institute, [3]Department of Biology, and [4]Quantitative Disease Ecology and Conservation Laboratory, Department of Geography, University of Florida, Gainesville, FL, USA
[5]Turner Enterprises Inc., 1123 Research Drive, Bozeman, MT, USA
[6]Department of Environmental Sciences, Policy, and Management, University of California, Berkeley, 130 Mulford Hall, Berkeley, CA, USA
[7]School of Mathematical Sciences, University of KwaZulu-Natal, Durban, South Africa

WMG, 0000-0001-8784-9354; SJR, 0000-0002-4308-6321; JKB, 0000-0003-0928-4831

Environmentally mediated indirect pathogen transmission is linked to host movement and foraging in areas where pathogens are maintained in the environment. In the case of anthrax, spores of the causative bacterium *Bacillus anthracis* are released into the environment following host death and create locally infectious zones (LIZs) around carcass sites; by grazing at LIZs, herbivores are potentially exposed to spores. Here, we used camera traps to assess how ungulate species use carcass sites in southwestern Montana and evaluated how these behaviours may promote indirect anthrax transmission, thus providing, to our knowledge, the first detailed documentation and study of the fine-scale mechanisms underlying foraging-based disease transmission in this ecosystem. We found that carcasses at LIZs significantly increased aboveground biomass of vegetation and concentrations of sodium and phosphorus, potentially making these sites more appealing to grazers. Host behavioural responses to LIZs varied depending on species, sex, season and carcass age; but, overall, our results demonstrated that carcasses or carcass sites serve as an attractant to herbivores in this system. Attraction to LIZs probably represents an increased risk of exposure to *B. anthracis* and, consequently, increased anthrax transmission rates. Accordingly, continued anthrax surveillance and control strategies are critical in this system.

# 1. Introduction

Anthrax, the disease caused by the Gram-positive, spore-forming bacterium *Bacillus anthracis*, is a zoonosis occurring nearly worldwide [1]. Instead of a typical transmission cycle in which disease is spread via direct contact between infected and susceptible individuals, *B. anthracis* spores persist in reservoirs and rely on environmentally mediated indirect transmission [2]. The posited common route of transmission for herbivores is ingestion of *B. anthracis* spores while grazing [3]. Once ingested, spores can germinate and proliferate, causing disease, and in many cases, host death. Following host death, the infective *B. anthracis* cells sporulate. Once formed, spores can remain in the soil for extended periods of time (potentially years to decades) until ingestion by another host, starting the cycle over again [4]. Accordingly, the carcass site and the area immediately surrounding the carcass of hosts killed by anthrax can become locally infectious zones (LIZs) where high concentrations of *B. anthracis* spores occur [5,6].

The deposition of animal carcasses, which seed LIZs, usually results in a pulse of nutrients into the environment [7–9]. These nutrient hotspots may support higher quality forage for herbivores, serving as an attractant in some ecosystems [7]. However, animals often exhibit behaviours in which they avoid areas potentially contaminated with parasites or pathogens. Rather than detect microparasites directly (especially bacteria and viruses), herbivores typically rely on parasite-associated signals. The presence of faeces is an important visual and olfactory cue of potential parasite presence [10], and selective foraging in the form of faecal avoidance has been well-documented in a variety of herbivore species [11–14]. Carcass sites hypothetically offer similar cues of possible pathogen presence, so they may be avoided by herbivores, regardless of the actual infection status of the carcass. This creates a trade-off for animals between grazing in areas with higher nutritive values and risking potential infection. Thus, LIZs could become preferred foraging areas, especially owing to a propensity for osteophagia in antler-growing cervids [15–17], or the lingering visual and olfactory cues could act as a deterrent to grazers. Avoidance or attraction to LIZs probably plays a crucial role in anthrax transmission and outbreaks.

Turner *et al.* [7] examined the interactions of herbivores with LIZs in Etosha National Park, Namibia; they found that while animal carcasses altered the environment, attraction to these sites varied among species of grazers and with carcass age. Though behaviours varied, generally grazing animals in the Etosha system shifted from avoidance to preference for LIZs over the first 2 years and eventual non-preference by 3 years. Those LIZs were confirmed positive with *B. anthracis* and may be a direct source of future infections [7].

Aside from studies in Etosha, how, where and when mammalian hosts contact *B. anthracis* in the environment has been poorly studied. Further, owing to its unique geography, findings from the subtropical, semi-arid system in Etosha are not applicable to many other endemic anthrax zones globally. In order to examine the response of herbivores to carcass-mediated nutrient hotspots in a montane ecosystem, we conducted a longitudinal study at carcass sites of plains bison (*Bison bison bison*) and elk (*Cervus canadensis*) from 2016 to 2018 in southwestern Montana. Carcass sites served as proxies for LIZs, but were not contaminated with *B. anthracis* because of stringent regulations about the disposal of infectious anthrax carcasses in Montana [18]. The objectives of the study were to assess the effect of LIZs on: (i) soil nutrient composition and grass biomass; (ii) the quality of soil nutrients, specifically in relation to potential *B. anthracis* spore survival; and (iii) bison and elk presence and behaviour using motion-detecting camera traps.

# 2. Material and methods

## 2.1. Study area and carcass site selection

The study was conducted on a privately owned ranch (approx. 300 km$^2$) in southwestern Montana (figure 1). The ranch is bordered by the Gallatin National Forest to the south, the Gallatin River to the east and the Madison River to the west. It belongs to the Greater Yellowstone Ecosystem and consists of 27% shrublands, 31% grasslands, 36% coniferous forests and 5% deciduous forests [19,20]. The southern half of the study area is dominated by steep, forested terrain while the eastern portion primarily consists of more gently sloped grassland and shrubland. Elevations range from 1334 to 3352 m. The climate is temperate, with cold, snowy winters and warm, dry summers; much of the ground is snow-covered from November through to March. The ranch manages domestic plains bison

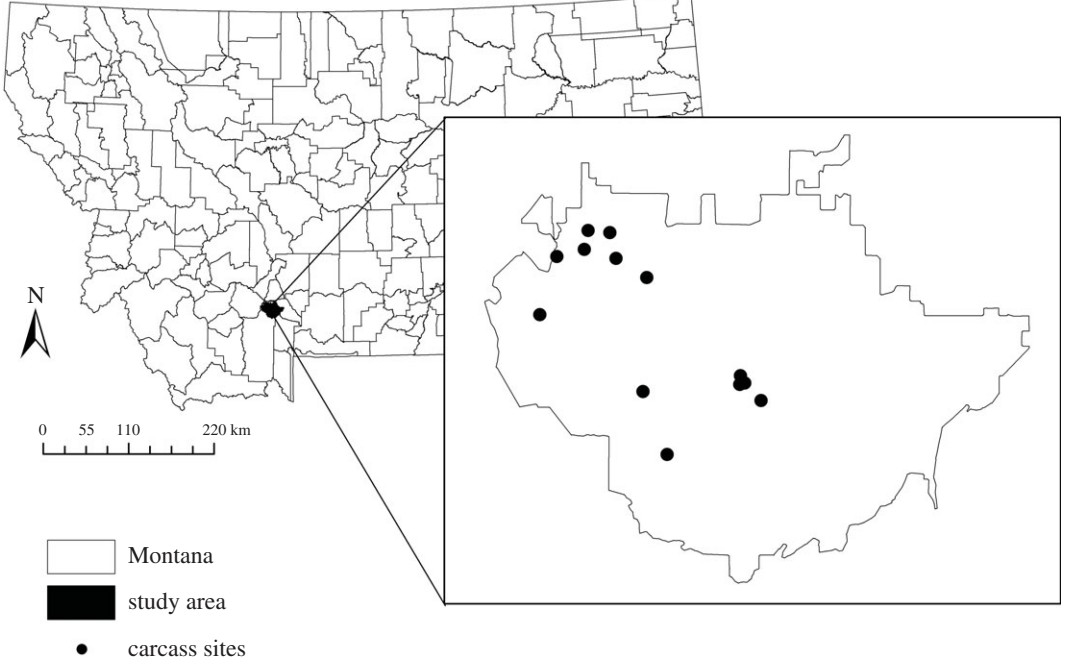

**Figure 1.** A map of the study site in southwestern Montana, USA, with points representing the locations of camera trap stations at 13 paired carcass sites (proxies for locally infectious zones; LIZs) and control sites.

as livestock while also promoting wildlife conservation. Fencing limits bison to the ranch itself but does not prohibit the movement of other species. Generally, bison pastures are large, with some pastures greater than 5000 ha, and bison can free range within pastures and between pastures when gates are open.

A major anthrax epizootic affected this ranch in July and August 2008, when at least 300 plains bison and 43 male elk succumbed to the disease [21]. Since 2008, routine surveillance has been conducted, and two bison found killed by wolves (*Canis lupus*) tested positive for active *B. anthracis* infection in 2010 [21]. Ongoing serological work at the site suggests that *B. anthracis* has continued to circulate since the 2008 outbreak [22]. Additionally, seroprevalence studies found that approximately 30% of male elk and 27% of unvaccinated bison tested were sero-positive for anthrax exposure (based on protective antigen; PAG [23]), suggesting that low-level exposure and survival of sub-lethal infections are relatively common in this area [22,23]. However, it is unknown when, where and how these hosts encounter *B. anthracis* in the environment.

For this study, we established camera trap stations at 13 LIZs and 13 paired control sites. We opportunistically used a combination of bison (10) and elk (3) carcasses, and motion-triggered camera traps (Reconyx Hyperfire PC800 Series, Browning Recon Force HD or Bushnell Trophy) were set up as close to the day of animal death as possible, usually within 24 h. LIZ camera traps were positioned with the carcass in view, and control cameras were paired approximately 20 m away from LIZ cameras, in a habitat with comparable soil, vegetation and elevation, oriented away from the carcass. Paired LIZ and control cameras were placed in the same pasture to control for the periodic movement of bison into and out of pastures by ranch managers. The location of each carcass was demarcated with a 3 × 3 m grid, the corners of which were marked with rocks or stakes. Cameras were placed approximately 10 m from the centre of the grid at a height of 1–1.5 m, and they were programmed to take photos without delay, at a rate of one picture every second once motion triggered. Monitoring began at five LIZs in the summer and autumn of 2016, one in the autumn of 2017 and seven in the spring of 2018. Cameras were visited weekly or biweekly to check battery levels and download photos from the external data cards.

## 2.2. Soil and vegetation collection

Vegetation and soil sampling were conducted in June 2018, which was approximately 21 months after carcass deposition for six sites and three months after deposition for seven sites. Adapting the

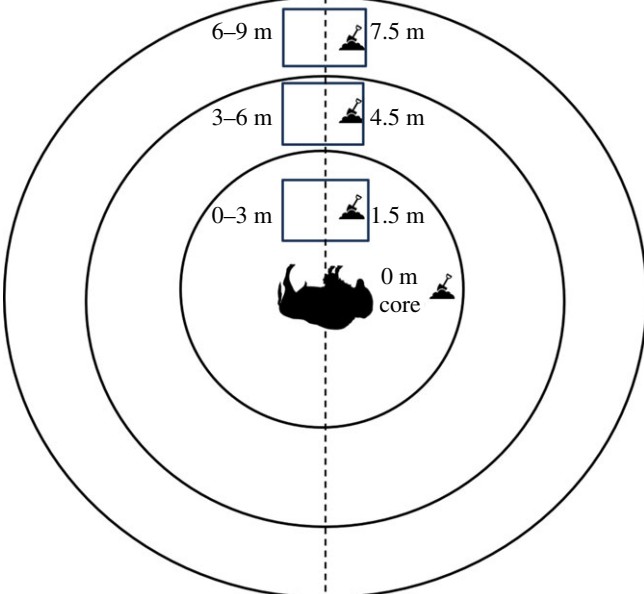

**Figure 2.** The sampling design for the soil and vegetation study. Soil cores were taken at 0 m, 1.5 m, 4.5 m and 7.5 m at each locally infectious zone (LIZ). There were three vegetation sampling zones, 0–3 m, 3–6 m and 6–9 m; aboveground biomass was taken from an 80 × 80 cm quadrat in each of these zones.

methods of Turner *et al.* [7], samples were collected from three sampling zones radiating outwards from the centre of the marked grid at LIZs: 0–3 m, 3–6 m and 6–9 m (figure 2). An 80 × 80 cm quadrat was placed in the centre of each zone along a transect. Within the quadrat, plant cover, density and maximum stem height were recorded. At control sites, two random quadrats were taken. The aboveground vegetation was harvested and weighed before drying in an Excalibur Food Dehydrator (Model 2900ECB; Excalibur) for 18 h at 40°C. After drying, the vegetation was weighed again for a measure of aboveground dry biomass. Soil cores of 10 cm in depth were taken in the centre of each sampling zone (at 0 m, 1.5 m, 4.5 m and 7.5 m). At control sites, two random soil samples were taken, unrelated to the sampling zone. Soil samples were analysed by the University of Florida's Institute of Food and Agricultural Sciences Extension Soil Testing Laboratory in Gainesville, Florida for percentage nitrogen (N), potassium (K), phosphorus (P), copper (Cu), sodium (Na), magnesium (Mg) and pH. The Kjeldahl method was used to assess the percentage of N in soil samples [24].

## 2.3. Camera trap analysis

Images were organized and analysed using the image management software DIGIKAM [25]. Each picture automatically recorded the date and time of capture; this information was used for both diurnal activity assessments and seasonality analyses. The summer months (June–August) were of particular interest because these months are the historical anthrax outbreak season in the region, dubbed the 'anthrax risk period' [20,21].

Each picture was manually assigned metadata tags classifying the number of animals in the photo as well as the species, age class, sex and behaviour. For ungulates, possible behaviour tag assignments were 'foraging', 'dominance', 'transit' or 'interacting with the carcass'. Foraging was indicated if an individual had its head at grass height and moving in a way that could signal biting, chewing or pulling at vegetation. Dominance behaviours were displays of individuals acting aggressively, including charging and locking antlers with other conspecifics or allospecifics. Transit was considered to occur if an animal walked, stood or sat in the study area without displaying foraging or dominance behaviours. Lastly, interaction with a carcass was deemed to occur if an individual smelled or touched a carcass with any part of its head but did not forage. Where the sex or age class of an animal was unable to be readily identified, these categories were assigned a tag of 'unknown'. All metadata was subsequently extracted from camera trap images using EXIFTOOL, a command-line application for

reading and editing meta information [26]; metadata was subsequently imported into R using the camtrapR package [27] (source code available at doi:10.5281/zenodo.3699573).

## 2.4. Soil nutrients and vegetation at locally infectious zones versus control sites

Statistical differences between the concentration of soil nutrients (mg kg$^{-1}$ of N, K, P, Cu, Na, Mg), pH and vegetation (dry biomass) at LIZs and control sites were analysed using Mann–Whitney $U$ tests [28]. For our Mann–Whitney tests, samples closest to the carcass at LIZs (vegetation in the 0–3 m zone, soil at 0 m) were compared to control samples. Sites were initially grouped for analysis by carcass age; however, testing determined that the six 21-month-old sites and the seven 3-month-old sites did not represent significantly different populations in nutrient content or vegetative biomass.

## 2.5. Distance decay of soil nutrients and vegetation at locally infectious zones

To assess factors affecting soil N, pH, K, P, Cu, Na and Mg, we fitted linear mixed models (LMMs) to concentration values for all nutrients and quantity of aboveground biomass. LMMs incorporate both fixed effects (variation explained by the independent variables of interest) and random effects (variation that is *not* explained by the independent variables of interest) [29]. As fixed effects, we included the following continuous variables: distance from the carcass (sampling zone); the age of the carcass in years; and the interaction between the sampling zone and carcass age. We controlled for variation among the different LIZs by using site as a random effect. Models were compared using Akaike's information criterion with correction for small sample sizes (AICc) to select the best candidate set of variables [30]. While the interaction between sampling zone and carcass age were initially included in all models, this interaction was excluded from final models if it did not significantly improve model fit by reducing the AICc value.

## 2.6. Ungulate use of locally infectious zones and control sites

### 2.6.1. Generalized linear models

Generalized linear models (GLMs) were employed to identify and characterize the effects of a variety of time and season-related predictor variables on the probability of grazing by ungulates. The set of predictors included the age of the carcass in days (at LIZs), the age of the carcass in years (at LIZs), the ordinal date (specific seasonality), the season (broad seasonality—winter, spring, summer and autumn) and the number of days since the start of the study to account for the effects of events in time that occur across sites (e.g. a drought). The probability of grazing was represented as a binary variable indicating if individuals at LIZs or control sites either grazed (a value of 1) or did not graze (a value of 0) at a site on a certain day. To relate grazing probability to the predictor variables, we employed a GLM with a binomial error and a logit link function. An initial set of GLMs grouped males and females of the same species together; this set of models was used to test for overall differences in grazing probability over time. The second set of models grouped bison males and females and elk males and females separately. AICc was used to compare GLMs.

### 2.6.2. Generalized additive models

To further analyse the relationships among site type, animal behaviour, time of year and time since animal death, we used a generalized additive model (GAM; [31]). As the name implies, GAM is an additive modelling technique where the impact of the predictive variables is captured through smoothing functions which, depending on the underlying patterns in the data, can be nonlinear [32]. Accordingly, GAMs can help identify patterns that may be missed with parametric models like GLMs. The basic structure of a GAM can be written as $g(E(Y)) = \alpha + f_1(x_1) + f_2(x_2) + K \ldots + f_p(x_p)$, where $Y$ is the dependent variable, $(E(Y))$ denotes the expected value and $g(Y)$ denotes the link function that links the expected value to the predictor variables $x_1, \ldots, x_p$. The terms $f_1(x_1), \ldots, f_p(x_p)$ denotes smooth, nonparametric functions.

Our GAM model, adapted from Turner *et al.*'s [7] Etosha National Park 5-variable model, is a seven-variable (i.e. $p = 7$ variables) model built to predict the probability of an animal grazing given its presence, written as $Y_t = G_t/N_t$ (ranging between 0 and 1). The ratio $Y_t$ is the sum of all individuals grazing per camera trap trigger ($G$) to the total number of individuals present per

camera trap trigger ($N$) in photos taken on day $t$, given that the study began at time $t = 0$. The model was of the form

$$P(G_t/N_t) = 1/(1 + e^{-\beta}),$$

where

$$\beta = a_s + f_1\left(\sqrt{N_t}\right) + f_2(t) + f_3(J|S) + f_4(J|C) + f_5(A_t|C) + f_6(J|Z) + f_7(A_t|Z) + \varepsilon_t,$$

and $\alpha$ is a constant, $\varepsilon$ is a quasi-binomial error term, and the response variable is assumed to depend on site $S$ ($S = 1, \ldots, 13$), conditioned on site type $C$ (LIZ or control), and on time $t$ through seasonal effects related to a continuous-time ordinal day variable $J$. The two extra variables in our model arise from our inclusion of the sex $Z$ of the grazing individuals in our analysis.

Using the notation $U|V$ to represent continuous variable $U$ conditioned on categorical variable $V$, the seven variables in our model are: $x_1 = (\sqrt{N_t})$ represents dependence on population density (the square-root transformation stabilizes the variance); $x_2 = t$ represents dependence on time; $x_3 = J|S$ represents the ordinal date depending on site; $x_4 = J|C$ represents the ordinal date depending on treatment (LIZ or control); $x_5 = A_t|C$ represents the days since animal death depending on the treatment, $C$ (i.e. 'age' of LIZ carcass); $x_6 = J|Z$ represents the ordinal date depending on animal sex, $Z$; $x_7 = A_t|Z$ represents the days since animal death depending on animal sex, $Z$.

In our analysis, we assumed quasi-binomial thin plate regression splines for all smoothing functions other than those involving ordinal dates. Thin plate splines are ideal for examining the combined effect of two or more continuous predictors [32], but in the case of ordinal date, cyclic cubic regression splines were used to ensure that the value of the smoother matched at days 1 and 365. The smoothing parameter was chosen based on the generalized cross validation (GCV) criterion within the mgcv package [33] in R. The GCV criterion compares the fit of all models based on all possible values of the smoothing parameter before choosing the one that fits best. Our source code and example dataset can be found at doi:10.5281/zenodo.3699573.

# 3. Results

## 3.1. Soil nutrients and vegetation at locally infectious zones versus control sites

We found that concentrations of P ($U = 41$, $p = 0.026$; figure 3$a$) and Na ($U = 23$, $p = 0.001$; figure 3$f$) were significantly higher at LIZs than at control sites. K ($U = 50$, $p = 0.08$; figure 3$c$) was typically higher at LIZs than at control sites but was only marginally significant. Na was the only nutrient to maintain higher concentrations at LIZs from the first year after animal death to 2 years post-mortem. The quantity of aboveground dry biomass was also significantly higher at LIZs ($U = 25$, $p = 0.002$; figure 3$h$). Concentrations of N, pH, Cu and Mg were not significantly different between LIZs and control sites.

## 3.2. Distance decay of soil nutrients and vegetation at locally infectious zones

We used LMMs to assess factors affecting soil P, N, K, Cu, Mg, Na, pH and biomass of aboveground vegetation. Our best LMMs included sampling zone and carcass age in years as fixed effects and site as the random effect for each nutrient. Model estimates indicate that soil pH increased with distance from the carcass centre (1.5 m: std. $\beta = 0.37$, s.e. $= 0.13$, $p < 0.01$; 4.5 m: std. $\beta = 0.46$, s.e. $= 0.13$, $p < 0.001$; electronic supplementary material, figure S1G), but the trend did not extend to all sampling zones, as the pH at 7.5 m was not significantly different from that at 4.5 m (7.5 m: $p = 0.06$). The pH was also significantly lower in the second year after animal death, an effect which was considered large (std. $\beta = -1.12$, s.e. $= 0.43$, $p < 0.05$). Concentrations of soil Na decreased significantly with distance from the carcass centre as well (1.5 m: std. $\beta = -1.24$, s.e. $= 0.34$, $p < 0.001$; 4.5 m: std. $\beta = -1.09$, s.e. $= 0.34$, $p < 0.01$; 7.5 m: std. $\beta = -1.25$, s.e. $= 0.34$, $p < 0.001$; electronic supplementary material, figure S1F). Additionally, the aboveground dry biomass significantly decreased further from the carcass (3–6 m: std. $\beta = -0.94$, s.e. $= 0.34$, $p < 0.01$; 6–9 m: std. $\beta = -0.90$, s.e. $= 0.34$, $p < 0.05$; electronic supplementary material, figure S1H). There were no detected effects of distance from the carcass site or carcass age for concentrations of K, Cu, Mg, P or N.

## 3.3. Ungulate use of locally infectious zones and control sites

From August 2016 to September 2018, camera traps were triggered 725 421 times. Of these, 116 228 pictures contained animals; 43 314 were of bison and 24 126 were of elk. Of bison captured by camera

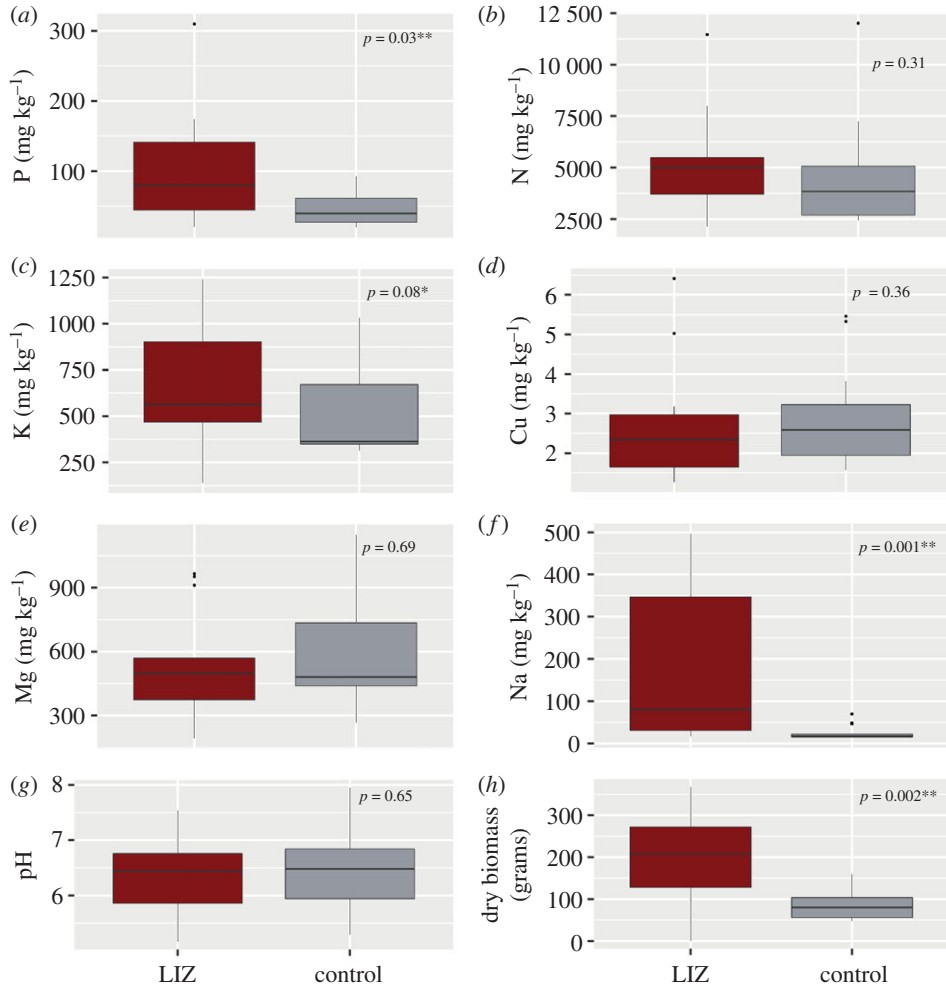

**Figure 3.** Results of Mann–Whitney $U$ tests comparing soil nutrient concentrations and quantity of aboveground biomass for LIZs and control sites. LIZ samples (red boxes) were collected from the sampling zone closest to the carcass, which was 0 m for soil samples and 0–3 m for vegetation quadrats.

traps, the majority were female (94% female at control sites, 88% female at LIZs). This sex discrepancy was expected owing to the larger numbers of female bison kept on the ranch. Bison of both sexes spent the most time in transit, accounting for 67% of the behaviour at control sites and 59% of the behaviour at LIZs. Thirty-one per cent of bison at LIZs were grazing females, which was similar to 33% of grazing females at controls. However, there was an increase in the proportion of males grazing between control sites and LIZs; less than 1% of pictures were of males grazing at control sites while nearly 4% were of males grazing at LIZs. The sex ratios were the opposite for elk, in which most animals captured were males (89% males at control sites, 65% males at LIZs). Elk of both sexes spent more time foraging at LIZs than at control sites—16% at control sites and 28% at LIZs for males and 4.7% at control sites and 18% at LIZs for females. Diurnal use patterns also showed differences between these two species, in which bison (electronic supplementary material, figure S2A) were primarily active during the daytime, from 9.00 to 19.00, while elk (electronic supplementary material, figure S2B) were mostly crepuscular (twilight-active), with the highest amount of activity from 4.00 to 8.00 and 19.00 to 23.00. These diurnal use patterns were expected based on the life-history characteristics of these two species and indicate that the presence of camera traps did not disturb the animals' regular behaviour patterns.

### 3.3.1. Generalized linear models

The first set of GLMs grouped all bison and all elk (regardless of sex) in order to examine the influence of time-related predictor variables on the probability of grazing for each species. The bison GLM with the lowest AICc value incorporated the predictor variables of season and carcass age in years. This model

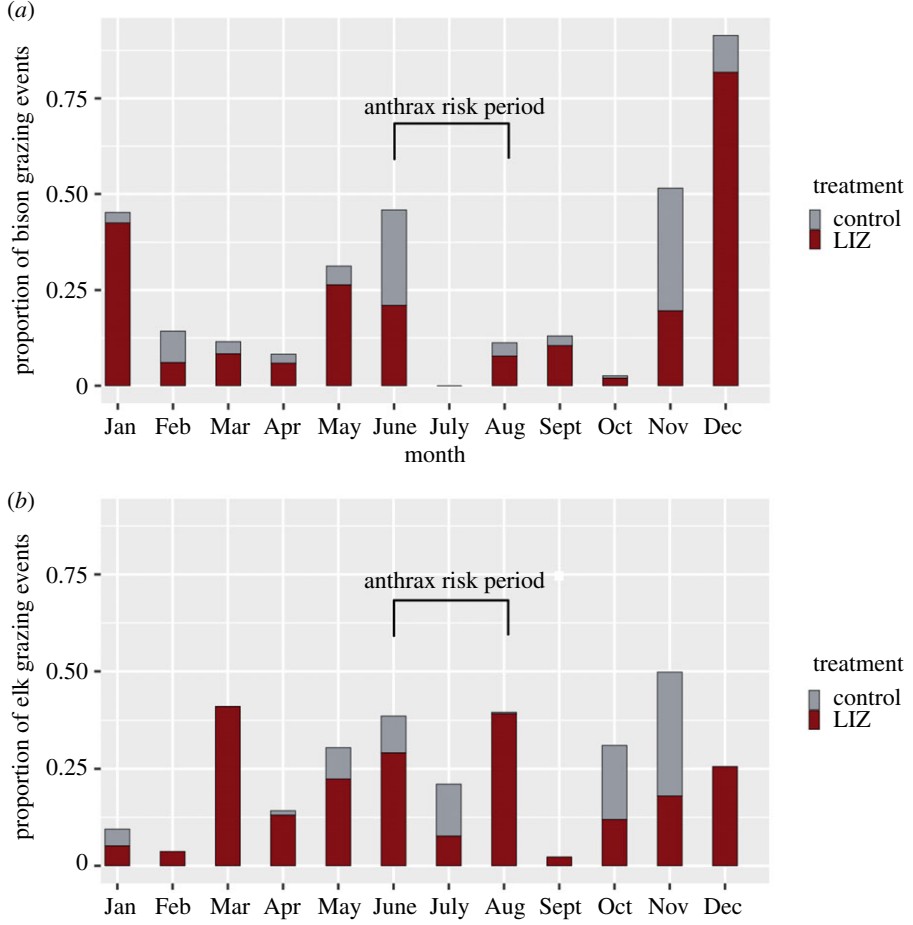

**Figure 4.** The seasonality of bison (*a*) and elk (*b*) grazing events at locally infectious zones (LIZs) and control sites. The proportion is the number of individuals grazing per camera trigger divided by the total number of individuals present on a given day, averaged over each month. Bison graze less at LIZs during the summer, while elk graze more at LIZs during spring and summer, which includes the anthrax risk period (June–August).

indicated that the probability of bison grazing decreased over the summer months, which corresponds to the anthrax risk period ($p < 0.0001$; figure 4*a*) and increased as the carcass aged ($p = 0.01$). The best-performing elk GLM incorporated the predictor variables of season, carcass age in years and days since the start of the study. By contrast with bison, the probability of elk grazing decreased as the carcass aged. Elk were also somewhat more likely to graze during the summer (anthrax risk period; marginally significant, $p = 0.06$; figure 4*b*) and spring (marginally significant, $p = 0.07$).

The second set of GLMs grouped males and females of each species separately and then further divided these groups by treatment (LIZs versus controls). For these GLMs, only the models predicting the probability of grazing for female bison at LIZs, male bison at LIZs, female bison at controls, male elk at LIZs and female elk at controls performed better than null models (table 1). For both female and male bison at LIZs, the best models predicting grazing included the age of the carcass in years, the age of the carcass in days, and the ordinal date. For both sexes the model-averaged coefficient estimate for ordinal date was slightly positive, indicating that bison were more likely to graze later in the year. The coefficient estimates for carcass age in years were also positive and significant for both sexes, indicating that the probability of grazing increased as the carcass aged. For female bison at control sites, the best model included the ordinal date, season and the number of days since the start of the study. The model coefficient for the season was negative, meaning that the probability of grazing was higher in winter (assigned a value of 1 in the analysis) than in the summer (seasonal value of 3) or autumn (seasonal value of 4). For male elk at LIZs, the best model included the carcass age in years and the number of days since the start of the study. The coefficient estimate for carcass age in years was negative indicating that male elk were more likely to graze at fresher carcasses. The best model for female elk at controls included only the days since the start of the study; the second-best model was the null.

**Table 1.** GLMs results of the grazing probability of female and male bison at locally infectious zones (LIZs), female bison at control sites, male elk at LIZs and female elk at control sites. (Predictors included carcass age in years and in days, days since the start of the study, ordinal date and season. Null models performed best for male bison at LIZs and control sites, male elk at LIZs and control sites, and female elk at LIZs. Significant $p$-values are in bold; significance codes: $*p < 0.05$; $**p < 0.01$; $***p < 0.001$.)

| response | best models | AICc | covariates | estimate | s.e. | z value | Pr(>\|z\|) |
|---|---|---|---|---|---|---|---|
| female bison at LIZs | 1. carcass age in days + carcass age in years + ordinal date | 143.0 | carcass age in days | −0.0104 | 0.0060 | −1.728 | 0.0839 |
| | | | carcass age in years | 5.2126 | 2.4717 | 2.109 | **0.0350**\* |
| | | | ordinal date | 0.0105 | 0.0029 | 3.681 | **0.0002**\*\*\* |
| | 2. carcass age in days + carcass age in years + ordinal date + days since start | 144.0 | carcass age in days | −0.0139 | 0.0072 | −1.941 | 0.0523 |
| | | | carcass age in years | 6.4912 | 2.8817 | 2.253 | **0.0243**\* |
| | | | ordinal date | 0.0104 | 0.0029 | 3.627 | **0.0002**\*\*\* |
| | | | days since start | 0.0012 | 0.0011 | 1.059 | 0.2898 |
| | 3. carcass age in years + ordinal date | 144.2 | carcass age in years | 1.0540 | 0.4711 | 2.237 | **0.0253**\* |
| | | | ordinal date | 0.0094 | 0.0027 | 3.550 | **0.0003**\*\*\* |
| male bison at LIZs | 1. carcass age in days + carcass age in years + ordinal date | 37.7 | carcass age in days | −0.028 | 0.0161 | −1.734 | 0.0829 |
| | | | carcass age in years | 13.653 | 6.9202 | 1.973 | **0.0485**\* |
| | | | ordinal date | 0.0142 | 0.0075 | 1.882 | 0.0598 |
| | 2. carcass age in days + carcass age in years + ordinal date + season | 38.2 | carcass age in days | −0.0559 | 0.0277 | −2.018 | **0.0436**\* |
| | | | carcass age in years | 25.137 | 11.7768 | 2.134 | **0.0328**\* |
| | | | ordinal date | 0.0263 | 0.0122 | 2.162 | **0.0306**\* |
| | | | season | −1.1147 | 0.7473 | −1.492 | 0.1358 |

(Continued.)

**Table 1.** (*Continued.*)

| response | best models | AICc | covariates | estimate | s.e. | z value | Pr(>\|z\|) |
|---|---|---|---|---|---|---|---|
| female bison at controls | 1. ordinal date + season + days since start | 139.5 | ordinal date | 0.0269 | 0.0094 | 2.859 | **0.0043**\*\* |
| | | | season | −2.0218 | 0.8077 | −3.134 | **0.0017**\*\* |
| | | | days since start | −0.0021 | 0.0010 | −1.993 | **0.0463**\* |
| | 2. ordinal date + season | 141.5 | ordinal date | 0.0206 | 0.0084 | 2.442 | **0.0146**\* |
| | | | season | −2.0218 | 0.7274 | −2.780 | **0.0054**\*\* |
| | 3. season | 146.6 | season | −0.3350 | 0.1816 | −1.845 | 0.0651 |
| male elk at LIZs | 1. carcass age in years + days since start | 164.7 | carcass age in years | −1.3628 | 0.5870 | −2.322 | **0.0203**\* |
| | | | days since start | 0.0034 | 0.0015 | 2.211 | **0.0270**\* |
| | 2. carcass age in years + days since start + season | 165.4 | carcass age in years | −1.3913 | 0.5875 | −1.155 | 0.2482 |
| | | | days since start | 0.0032 | 0.0015 | −2.368 | **0.0179**\* |
| | | | season | −0.2495 | 0.2161 | 2.105 | **0.0353**\* |
| | 3. carcass age in days + days since start | 166.2 | carcass age in days | −0.0033 | 0.0016 | 2.081 | **0.0375**\* |
| | | | days since start | 0.0034 | 0.0016 | −2.067 | **0.0387**\* |
| female elk at controls | 1. days since start | 24.8 | days since start | 0.0099 | 0.0091 | 1.090 | 0.276 |

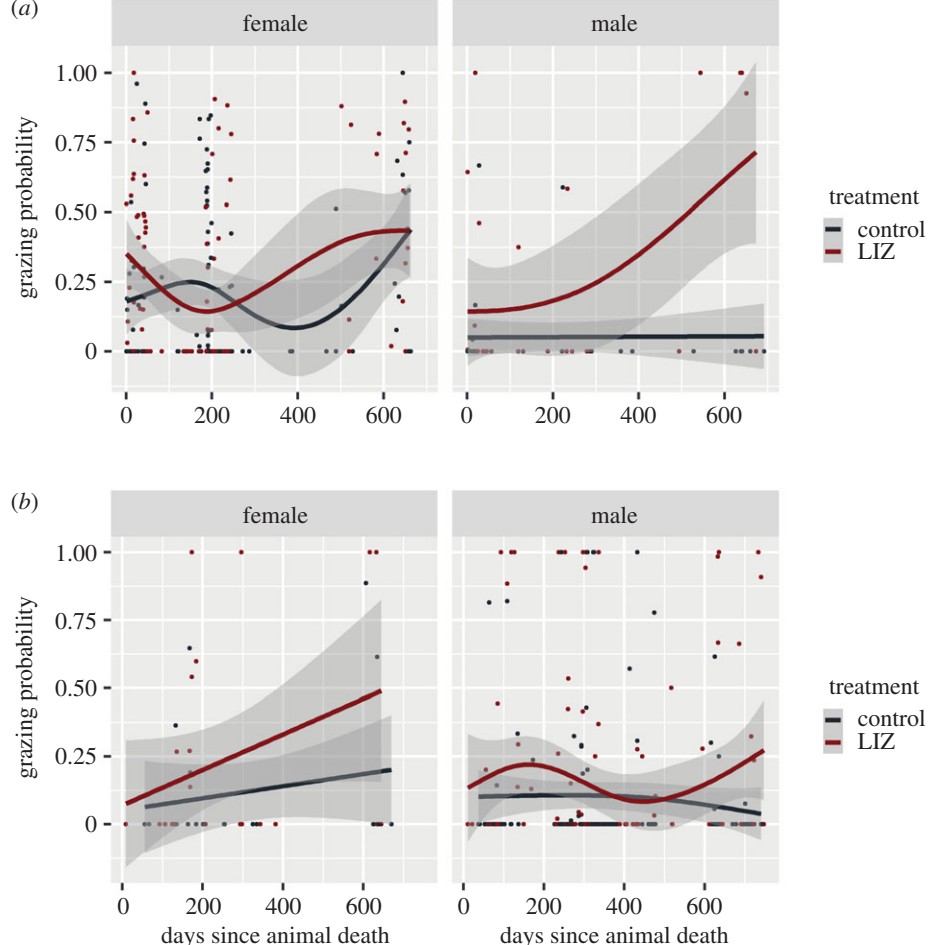

**Figure 5.** Results of GAMs showing the probability of an animal grazing given that the animal is present for locally infectious zones (LIZs) versus control sites over time since animal death for bison (*a*) and elk (*b*).

### 3.3.2. Generalized additive models

GAMs indicated that female bison initially showed a decrease in grazing probability at LIZs and then a gradual increase as the carcass aged, from around 200 days onwards (figure 5*a*). The probability of male bison grazing at controls remained nearly constant but increased as the carcass aged for LIZs. Grazing probability for female elk were similar to male bison, as they were more likely to graze at LIZs as the carcass aged (figure 5*b*). Male elk were more likely to graze at LIZs in the first year after animal death, but their probability of grazing decreased in the second year after animal death.

## 4. Discussion

In this study, we examined the effect of proxy-LIZs on the environment and on the behaviour of bison and elk to assess the potential risks for foraging-based disease transmission in a montane ecosystem. To evaluate these topics, we employed several statistical techniques. These are summarized in table 2.

In determining if the soil nutrient composition and vegetation differed between LIZs and control sites, we found evidence that concentrations of some nutrients, most notably P and Na, were significantly higher at LIZs. The same was true for aboveground biomass of vegetation. K was typically higher at LIZs but was only marginally significant. N, Cu, Mg and pH showed little difference between LIZs and control sites. These results are not entirely novel, as previous studies of ungulate carcass effects have shown that they create a significant and localized pulse of nutrients into the environment [7–9,34]. However, we were particularly interested to compare our results in a temperate montane ecosystem to the previous Turner *et al.* [7] work in Namibia. In that system, the authors found that soil P, N, Na and K all had higher concentrations directly underneath zebra carcasses. Grass biomass was also significantly higher at the inner carcass zone, but that trend only

**Table 2.** Summary of statistical analyses used in this study.

| hypothesis | analytical approach | general findings |
| --- | --- | --- |
| soil nutrients and vegetation biomass are higher at LIZs than at control sites | mann–Whitney *U* tests | concentrations of P, Na and vegetation biomass were significantly higher at LIZs than at control sites |
| soil nutrients and vegetation biomass decrease radiating outwards from LIZ carcasses and decrease over time | linear mixed models (LMMs) | concentrations of Na and vegetation biomass significantly decreased with distance from the carcass centre, while pH significantly increased with distance. pH was significantly lower in the second year after animal death |
| ungulates are more likely to graze at LIZs as the carcass site ages and during the anthrax risk period (summer) | generalized linear models (GLMs) | bison are more likely to graze as the carcass site ages, but the opposite is true for elk. Bison were less likely to graze at LIZs during the summer |
| male bison and male elk are more likely to graze at LIZs than at control sites, and the probability of grazing at LIZs will not change as the carcass ages. Female bison and female elk are more likely to graze at LIZs as the carcass ages | generalized additive models (GAMs) | female elk and male bison are more likely to graze at LIZs than at control sites regardless of carcass age, and the likelihood of grazing at LIZs increases over time. Patterns of grazing at LIZs for male elk and female bison are more nuanced and change as the carcass ages |

held for 1 year after animal death. Thus, results in Etosha National Park seem to be similar to those in Montana, but perhaps not as long-lasting; six of the sites included in our study were sampled for soil nutrients and vegetation 21 months after animal death, and the trends of higher concentrations of nutrients and biomass still held. Consequently, if animals in temperate, more heterogenous landscapes like Montana are actively seeking higher quality forage, they may be attracted to carcass sites and continue returning to them for longer time scales than in Etosha. This has direct implications for anthrax transmission, as continued visitation to carcass sites increases opportunities for possible infection and subsequently, outbreaks.

Additionally, it is worth noting that our carcass sites acted as proxies for LIZs. Owing to the exsanguination that occurs at actual anthrax carcasses [7,35,36], they probably could lead to a greater localized nutrient release and wider dispersion of nutrients than from the non-anthrax carcasses used here, where blood coagulates in tissues. The nutrient pulse demonstrated at our non-infectious sites may be even greater in the case of an actual anthrax outbreak.

Of the nutrients increased at LIZs, higher Na concentrations could be particularly important for ungulates. Terrestrial plants contain around $1.0 \, \text{mg kg}^{-1}$ Na [37], whereas herbivores that eat them maintain Na levels 100 to 1000-fold higher [38]. Thus, it logically follows that significantly higher concentrations of Na exist at carcass sites. However, it is also important to note that Na inputs from rainfall decrease 1000-fold with distance inland and melting snowfall in mountains and high plains areas often leaches Na from the ground [39], which means this ecosystem may be extremely Na-limited for consumers. This, compounded by the fact that herbivores have been shown to select Na-rich forage [40–43], could be driving bison and elk to graze at LIZs, where Na is more readily available.

A secondary aim of our study was to determine if soil conditions for pathogen survival, specifically *B. anthracis*, were better at LIZs than at control sites. *Bacillus anthracis* spores concentrate in soils at anthrax carcass sites and can persist there for years, but spore survival depends on soil conditions [4,44]. There is consensus that spores survive best in slightly alkaline environments [2,7,44–49]. Here, the results of LMMs showed that pH was lowest in the inner zone of carcass sites and increased with distance from the carcass centre (electronic supplementary material, figure S1). However, pH values even at the

centre of LIZs were slightly alkaline (mean pH = 6.8), so values were suitable for *B. anthracis* throughout the LIZ, and pH is probably not a limiting factor in this system. These results were expected, as previous soil surveys of the study area have yielded an average soil pH of at least 7 [50].

Apart from examining soil and nutrient conditions, we also wanted to determine the potential importance of LIZs for foraging-based disease transmission in grazing herbivores in this ecosystem. We found that both elk and bison males and females grazed at carcass sites but behaved differently seasonally and as the carcasses aged. Bison grazed more at LIZs during the winter months (figure 4*a*). Owing to the cold winters and high yearly snowfall in southwest Montana (average snow depth from 2002 to 2019 at Lone Mountain, MT = 160 cm), this ecosystem is probably biomass limited in winter. Because the amount of aboveground biomass was significantly higher at LIZs compared to control sites, and LMMs also showed that biomass decreased with distance from the carcass (electronic supplementary material, figure S1), LIZs could attract bison to graze there in the winter, when finding other vegetation above the snowpack is difficult. Bison may be facing a trade-off between forage quality and potential pathogen exposure and choose pathogen exposure when resources are limited. Furthermore, because anthrax outbreaks primarily occur in the summer months, bison contacting *B. anthracis* during the winter could explain the high seroprevalence (approx. 27%) of unvaccinated bison for anthrax exposure [22] but rare yearly outbreaks in this system.

At our study site, the total elk population is around 1800 individuals, with roughly 1500 females and a few hundred males. Despite this, most elk caught by camera traps were male (88% male at control sites, 65% male at LIZs), possibly indicating a hierarchy exists in which dominant male elk continually revisit the same sites. Additional support for this theory comes from the fact that male elk showed dominance behaviours at LIZs but not at control sites, which could indicate that these individuals were competing over LIZs because of higher quality forage in these areas. At control sites, 18.11% of the camera trap images of male elk were of individuals grazing; at LIZs, that proportion was substantially higher: 43.82%. Female elk were also found to graze more at LIZs than at control sites, but the difference was less pronounced than in male elk. Male elk may be attracted to LIZs because of higher quality forage or their increased concentration of P, a vital nutrient for antler growth [15–17].

Additionally, elk of both sexes grazed more at LIZs during the spring and summer months (figure 4*b*). The summer (from June to August) is when anthrax outbreaks typically occur in this system, termed the 'anthrax risk period' [20,21]. Accordingly, elk preferences for grazing at LIZs during this time of year could be particularly important for the transmission of *B. anthracis* and potential anthrax outbreaks. The outbreak in 2008 had a high mortality rate for male elk, and serological exposure is higher in males in this system [20]. Both facts are supported by the behaviours of males documented here. Comparisons between male and female bison and elk are critical to evaluating anthrax transmission risk and are an expansion upon previous work by Turner *et al.* [7], where sex was not evaluated.

Anthrax control strategies can be grouped into three categories: (i) control strategies before the outbreak; (ii) control strategies during the outbreak; and (iii) control strategies after the impact of the outbreak [51]. Methods of control before an outbreak include surveillance, vaccination and movement control of animals out of high-risk areas. Owing to the high use and frequency of grazing at LIZs by bison and elk, the bison vaccination programme that is currently ongoing at this site [21] is crucial to continue. Control strategies during the outbreak include vaccination and quarantine. Results from this study show that both sexes of elk and bison use LIZs immediately after animal death; thus, practicing quarantine to keep susceptible individuals out of potential LIZs in the middle of an outbreak may be critical. Last of all, control strategies after an outbreak are proper disposal of animal carcasses, ideally through burning and/or deep burial, and then chemical surface decontamination. Our results indicate proper disposal of carcasses is also vital, owing to the frequent use of carcass sites by both elk and bison over long time scales.

Anthrax outbreaks have the potential to affect a range of wildlife and livestock species and could incur huge economic losses. Further, owing to its potential to infect humans, the disease presents a public health concern. Here, we show that there are subtle and critical indicators, arrived at through extensive and detailed camera trap monitoring, that can maximize the effectiveness of anthrax control before, during and after an outbreak. Our findings are applicable to future research and to help inform management decisions to prevent future outbreaks in a complex disease system.

Ethics. Camera trap data were analysed under an observational IACUC approved by the University of Florida (no. 201910563 to J.K.B.).

Data accessibility. All data for this study are available within the manuscript or in an online Zenodo repository, including the R code for performing analyses at doi:10.5281/zenodo.3699573.

Authors' contributions. S.J.R., J.M.P. and J.K.B. developed the study and W.M.G. revised the manuscript. V.A. and M.A.W. collected the data. M.A.W. analysed data and wrote the manuscript. M.U. analysed the data. All authors gave final approval for publication.

Competing interests. We declare we have no competing interests.

Funding. This work was supported by the National Institute of General Medical Sciences (NIGMS) of the National Institutes of Health (NIH) under award no. R01GM117617. Additional support was provided by the Emerging Pathogens Institute at the University of Florida.

Acknowledgements. We thank the ranch staff for access, logistical support, local knowledge and animal/data collection expertize. D. Zincke facilitated sample and data shipments. M. Norris and T. Hadfield provided insights on *B. anthracis* biology.

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
