## [Reviewer comments · Royal Society Open Science]

Review History

RSOS-200246.R0 (Original submission)

Review form: Reviewer 1

Is the manuscript scientifically sound in its present form?

No

Are the interpretations and conclusions justified by the results?

No

Is the language acceptable?

Yes

Do you have any ethical concerns with this paper?

No

Have you any concerns about statistical analyses in this paper?

Yes

Recommendation?

Major revision is needed (please make suggestions in comments)

Comments to the Author(s)

See attached (Appendix A).

Review form: Reviewer 2

Is the manuscript scientifically sound in its present form?

Yes

Are the interpretations and conclusions justified by the results?

Yes

Is the language acceptable?

Yes

Do you have any ethical concerns with this paper?

No

Have you any concerns about statistical analyses in this paper?

No

Recommendation?

Accept with minor revision (please list in comments)

Comments to the Author(s)

Authors are studying 13 localities of a southwestern ranch in Montana, USA to show how minerals and vegetation have higher concentrations on sites with carcasses than areas without. Authors chose this area of study for being a previous focus of an anthrax outbreak despite the lack of current anthrax cases due to control strategies. Thus, they are showing a proxy of anthrax local suitability around non-infected carcasses. The manuscript is well written and evidence is well presented. A couple of comments that should be definitely address:

Although a map is provided. I strongly recommend adding decimal coordinates of the localities studied, as they can be helpful to future spatial-oriented studies.

Line 59-60: Indicate the reference of the landscape characteristics defining the study area, that is, the scrubland, grasslands, coniferous forest, and deciduous forests categorization.

Line 181: Please add a space after the symbol of 'Error term' in the equation.

Line 210: Sometimes you use the word to describe the soil elements and others the symbol (Phosphorus vs P) use either one or the other and be consistent across the manuscript.

Line 211: Why potassium is reported as statistically significant with a $p = 0.08$? If a threshold of significance different than 0.05 was established, this should be stated in the methods section. Otherwise this should be non-significant. Why the value reported here is W instead of U as in the other cases?

Supplementary Figure 1 legend: Consider adding letters to each panel, in the main text you are referring to supplementary figure 1G which lacks the 'G' for identification (line 225; see also line

229 referring to supp. fig. 1F). In line 225, referring to the same figure, notice that the distance of 7.5 m is not statistically significant ($p = 0.06$) but it is not mentioned in the main text.

Anthrax risk period as depicted in Figure five has never been mentioned or referenced (see line 395) in the manuscript. It will be important to reference and comment this specific time interval for interpretation purposes.

Figure 3: In the legend please specify that left box plots (or red box plots) correspond to the local infectious zones. Notice that in the first column the x axis for LIZ is named as 0 m and in the second column as 0-3 without unit. Please use the same labels for both columns (e.g., LIZ vs Control) and add any extra detail (0-3 m for biomass quadrat) in the legend.

Line 338: Please add a reference to support the exsanguination of anthrax carcasses.

Line 359: This should be supplementary figure 1. Please revise.

Line 373: Supplementary figure 1. Please revise.

Line 395: Add a reference for the 'anthrax risk period'

I also would like to encourage to add any of the 'cover' pictures as part of the manuscript main text.

Decision letter (RSOS-200246.R0)

Dear Dr Blackburn,

The editors assigned to your paper ("Ungulate Use of Locally Infectious Zones (LIZs) in a Re-Emerging Anthrax Risk Area") have now received comments from reviewers. We would like you to revise your paper in accordance with the referee and Associate Editor suggestions which can be found below (not including confidential reports to the Editor). Please note this decision does not guarantee eventual acceptance.

Please submit a copy of your revised paper before 24-May-2020. Please note that the revision deadline will expire at 00.00am on this date. If we do not hear from you within this time then it will be assumed that the paper has been withdrawn. In exceptional circumstances, extensions may be possible if agreed with the Editorial Office in advance. We do not allow multiple rounds of revision so we urge you to make every effort to fully address all of the comments at this stage. If deemed necessary by the Editors, your manuscript will be sent back to one or more of the original reviewers for assessment. If the original reviewers are not available, we may invite new reviewers.

When submitting your revised manuscript, you must respond to the comments made by the

referees and upload a file "Response to Referees" in "Section 6 - File Upload". Please use this to document how you have responded to the comments, and the adjustments you have made. In order to expedite the processing of the revised manuscript, please be as specific as possible in your response.

- Data accessibility

If you wish to submit your supporting data or code to Dryad (<http://datadryad.org/>), or modify your current submission to dryad, please use the following link:
<http://datadryad.org/submit?journalID=RSOS&manu=RSOS-200246>

- Competing interests

- Authors' contributions

- Acknowledgements

- Funding statement

Associate Editor's comments:

The reviewers offer a number of useful comments on your work, and it seems to be on the right track for eventual acceptance; however, further consideration will be based on your satisfactorily responding to the concerns and queries raised. Please carefully work through the reviewers' comments in your revised paper. Good luck and we'll look forward to receiving this from you soon.

Comments to Author:

Reviewers' Comments to Author:
Reviewer: 1

Comments to the Author(s)
See attached (Ungulate use of locally infectious zones in a re-emerging Anthrax risk area.pdf)

Reviewer: 2

Comments to the Author(s)
Authors are studying 13 localities of a southwestern ranch in Montana, USA to show how minerals and vegetation have higher concentrations on sites with carcasses than areas without. Authors chose this area of study for being a previous focus of an anthrax outbreak despite the lack of current anthrax cases due to control strategies. Thus, they are showing a proxy of anthrax local suitability around non-infected carcasses. The manuscript is well written and evidence is well presented. A couple of comments that should be definitely address:

Although a map is provided. I strongly recommend adding decimal coordinates of the localities studied, as they can be helpful to future spatial-oriented studies.

Line 59-60: Indicate the reference of the landscape characteristics defining the study area, that is, the scrubland, grasslands, coniferous forest, and deciduous forests categorization.

Line 181: Please add a space after the symbol of 'Error term' in the equation.

Line 210: Sometimes you use the word to describe the soil elements and others the symbol (Phosphorus vs P) use either one or the other and be consistent across the manuscript.

Line 211: Why potassium is reported as statistically significant with a $p = 0.08$? If a threshold of significance different than 0.05 was established, this should be stated in the methods section. Otherwise this should be non-significant. Why the value reported here is W instead of U as in the other cases?

Supplementary Figure 1 legend: Consider adding letters to each panel, in the main text you are referring to supplementary figure 1G which lacks the 'G' for identification (line 225; see also line 229 referring to supp. fig. 1F). In line 225, referring to the same figure, notice that the distance of 7.5 m is not statistically significant ($p = 0.06$) but it is not mentioned in the main text.

Anthrax risk period as depicted in Figure five has never been mentioned or referenced (see line 395) in the manuscript. It will be important to reference and comment this specific time interval for interpretation purposes.

Figure 3: In the legend please specify that left box plots (or red box plots) correspond to the local infectious zones. Notice that in the first column the x axis for LIZ is named as 0 m and in the second column as 0-3 without unit. Please use the same labels for both columns (e.g., LIZ vs Control) and add any extra detail (0-3 m for biomass quadrat) in the legend.

Line 338: Please add a reference to support the exsanguination of anthrax carcasses.

Line 359: This should be supplementary figure 1. Please revise.

Line 373: Supplementary figure 1. Please revise.

Line 395: Add a reference for the 'anthrax risk period'

I also would like to encourage to add any of the 'cover' pictures as part of the manuscript main text.

Author's Response to Decision Letter for (RSOS-200246.R0)

See Appendix B.

RSOS-200246.R1 (Revision)

Review form: Reviewer 2

Is the manuscript scientifically sound in its present form?

Yes

Are the interpretations and conclusions justified by the results?

Yes

Is the language acceptable?

Yes

Do you have any ethical concerns with this paper?

No

Have you any concerns about statistical analyses in this paper?

No

Recommendation?

Accept with minor revision (please list in comments)

Comments to the Author(s)

All my comments have been addressed.

Last comment:

Line 376: the word 'that' is duplicated.

Decision letter (RSOS-200246.R1)

Dear Dr Blackburn

Please accept our apologies for the unusual delay in completing this round of review. On behalf of the Editors, we are pleased to inform you that your Manuscript RSOS-200246.R1 "Ungulate Use of Locally Infectious Zones (LIZs) in a Re-Emerging Anthrax Risk Area" has been accepted for publication in Royal Society Open Science subject to minor revision in accordance with the referees' reports. Please find the referees' comments along with any feedback from the Editors below my signature.

Please submit your revised manuscript and required files (see below) no later than 7 days from today's (ie 27-Aug-2020) date. Note: the ScholarOne system will 'lock' if submission of the revision is attempted 7 or more days after the deadline. If you do not think you will be able to meet this deadline please contact the editorial office immediately.

Kind regards,

Andrew Dunn

Reviewer comments to Author:

Reviewer: 2

Comments to the Author(s)

All my comments have been addressed.

Last comment:

Line 376: the word 'that' is duplicated.

===PREPARING YOUR MANUSCRIPT===

- one version identifying all the changes that have been made (for instance, in coloured highlight, in bold text, or tracked changes);
- a 'clean' version of the new manuscript that incorporates the changes made, but does not highlight them.

This version will be used for typesetting.

===PREPARING YOUR REVISION IN SCHOLARONE===

- 1) One version identifying all the changes that have been made (for instance, in coloured highlight, in bold text, or tracked changes);
 - 2) A 'clean' version of the new manuscript that incorporates the changes made, but does not highlight them.
 - An individual file of each figure (EPS or print-quality PDF preferred [either format should be produced directly from original creation package], or original software format).
 - An editable file of each table (.doc, .docx, .xls, .xlsx, or .csv).
 - An editable file of all figure and table captions.
- Note: you may upload the figure, table, and caption files in a single Zip folder.
- Any electronic supplementary material (ESM).
 - If you are requesting a discretionary waiver for the article processing charge, the waiver form must be included at this step.
 - If you are providing image files for potential cover images, please upload these at this step, and inform the editorial office you have done so. You must hold the copyright to any image provided.
 - A copy of your point-by-point response to referees and Editors. This will expedite the preparation of your proof.

- Ensure that your data access statement meets the requirements at <https://royalsociety.org/journals/authors/author-guidelines/#data>. You should ensure that you cite the dataset in your reference list. If you have deposited data etc in the Dryad repository, please only include the 'For publication' link at this stage. You should remove the 'For review' link.
- If you are requesting an article processing charge waiver, you must select the relevant waiver option (if requesting a discretionary waiver, the form should have been uploaded at Step 3 'File upload' above).
- If you have uploaded ESM files, please ensure you follow the guidance at <https://royalsociety.org/journals/authors/author-guidelines/#supplementary-material> to include a suitable title and informative caption. An example of appropriate titling and captioning may be found at https://figshare.com/articles/Table_S2_from_Is_there_a_trade-off_between_peak_performance_and_performance_breadth_across_temperatures_for_aerobic_scope_in_teleost_fishes_/3843624.

Author's Response to Decision Letter for (RSOS-200246.R1)

See Appendix C.

Decision letter (RSOS-200246.R2)

Dear Dr Blackburn,

It is a pleasure to accept your manuscript entitled "Ungulate Use of Locally Infectious Zones (LIZs) in a Re-Emerging Anthrax Risk Area" in its current form for publication in Royal Society Open Science. The comments of the reviewer(s) who reviewed your manuscript are included at the foot of this letter.

Appendix A

Ungulate use of locally infectious zones in a re-emerging anthrax risk area

The authors used a series of linear and additive models to quantify the relationship between animal use of carcass sites (proxy LIZs) and various covariates. I think the manuscript provides a solid foundation. However, I think the Methods and Results sections need to be improved so readers clearly understand the analyses that the authors performed. I think there may be some issues with terminology (general linear models v generalized linear models) as well.

The Methods section that describes the GAMs need some additional clarification. For example, this section never explicitly clarifies the dependent variable for the models. It appears from the results and R script provided that there the authors actually ran two GAM models: one that examined the number of visitations and the other model looking at the probability of grazing. I think using the same terminology and language for the dependent variables in the Methods and the Results is essential. The authors could also write out the equations more specifically rather than writing a general example as they did in line 181. What is the $P(Y)$? Is it $P(Y = 1)$? I would like to provide additional detailed recommendations, but it is difficult to provide these given it is tough to tell what the authors actually did.

Alternatively, I recommend the authors consider two parametric approaches to replace the GAMs. For the number of visitation, a Poisson model would suffice: $\log(\text{number of visitations}) = B_0 + B_1 * X_1, \dots \text{etc.}$

For the number of grazing events, a Poisson model with an offset term would be easier to interpret:

$\log(\text{number of grazing events}) = \log(\text{total herbivores present}) + B_0 + B_1 * X_1, \dots \text{etc.},$

The offset term means that you would effectively be modeling $\log(\text{number of grazing events} / \text{total herbivores present})$ as the dependent variable.

Specific comments

Lines 10 – 11: What is an “extended period of time”? Hours, days, months?

Lines 38 – 47: I think this is an extremely interesting question and the authors lay it out really well in the introduction. However, I do think 1 – 2 additional sentences are necessary to clarify an issue the study design. The authors hypothesize (based on a wealth of literature) that animals may avoid LIZs because of cues associated with a carcass in of itself. I have no background in this field, but is it possible that animals could detect the pathogen itself and possibly avoid real LIZs but be attracted to “proxay LIZs”? Obviously, there is no responsible way to test this in the field, but it seems the question should be posed somewhere.

Methods

Line 143: List the specific variables in parentheses that were tested with the Mann-Whitney U tests.

Line 152: What was the assumed distribution for the response variable in the generalized linear mixed models and why log transform these variables unless the authors assumed a normal distribution? If they assumed a normal distribution, then the models are not “generalized linear models” – they are just “linear mixed models” or “general linear mixed models”.

Line 165: Why were logistic regressions used to test for correlations? Why not use correlation metrics?

Line 170: If by Julian Day they mean a numerical day of the year from 1 to 365, then it should be called ordinal day.

Line 181: This could be written like this if they are talking about a logistic model:

$$\text{logit}(p) = \alpha + f_1(x_1) + f_2(x_2) + \dots + f(x_p) + \epsilon$$

where p represents the probability that...blah blah blah.

Line 191: Is N_i the number of animals that appear in a photo per day?

Results

The authors tested N, pH, **K**, **P**, Cu, **Na**, and Mg according to the Methods. A quick mention in this section that N, pH, Cu, and Mg were not significant would provide some clarity.

Lines 225, 229, and 232: Include parameter estimates for Na and pH.

Lines 256 – 260: A negative binomial model was used to examine the number of grazing events for each species. I don't remember this in the Methods and I see no other mention of “negative binomial” and “number of grazing events” in the Methods. This is a great example of using consistent terminology for dependent variables between the Methods and Results.

Lines 258 – 259: Is this the first mention that the anthrax risk period is June – August?

Line 261: I would replace “generalized linear model” with mixed logistic regression where appropriate. It defines the model with fewer words.

Lines 297 – 299: What are the authors talking about here with “host death”?

Appendix B

Response to Reviewers

Overview:

Apart from some minor edits, the major takeaway from the reviewers' comments was that the statistical methodology was confusing and somewhat cumbersome to understand. To address these critiques, we have changed some labeling and used more concise, uniform, and intentional statistical language throughout the manuscript. We have also made a concerted effort to expound on the methodology by adding more in-depth background information for the more complex methods employed (LMMs and GAMs) and clarifying the purpose of using these statistical techniques. Further, we have added a methodology table that we believe will help readers quickly identify the hypotheses, reasoning, and general conclusions of each method.

Our responses are in blue throughout this letter.

Reviewers' Comments to Author:

Reviewer: 1

Comments to the Author(s)

The authors used a series of linear and additive models to quantify the relationship between animal use of carcass sites (proxy LIZs) and various covariates. I think the manuscript provides a solid foundation. However, I think the Methods and Results sections need to be improved so readers clearly understand the analyses that the authors performed. I think there may be some issues with terminology (general linear models v generalized linear models) as well.

The Methods section that describes the GAMs need some additional clarification. For example, this section never explicitly clarifies the dependent variable for the models. It appears from the results and R script provided that there the authors actually ran two GAM models: one that examined the number of visitations and the other model looking at the probability of grazing. I think using the same terminology and language for the dependent variables in the Methods and the Results is essential. The authors could also write out the equations more specifically rather than writing a general example as they did in line 181. What is the $P(Y)$? Is it $P(Y = 1)$? I would like to provide additional detailed recommendations, but it is difficult to provide these given it is tough to tell what the authors actually did.

Alternatively, I recommend the authors consider two parametric approaches to replace the GAMs. For the number of visitation, a Poisson model would suffice: $\log(\text{number of visitations}) = B_0 + B_1 \cdot X_1, \dots \text{etc.}$

For the number of grazing events, a Poisson model with an offset term would be easier to interpret:

$\log(\text{number of grazing events}) = \log(\text{total herbivores present}) + B_0 + B_1 * X_1, \dots \text{etc.},$

The offset term means that you would effectively be modeling $\log(\text{number of grazing events} / \text{total herbivores present})$ as the dependent variable.

- We agree with the reviewer that the GAM section was confusing and needed clarification. Accordingly, the GAM methods have been heavily revised and now include specific example equations and more detail.
- The reviewer is correct that we initially ran two GAM models – one in which the dependent variable is the probability of visitation and the other in which the dependent variable is the probability of grazing given that an individual is present. However, only the GAMs for the probability of grazing were included here, as grazing, rather than visitation alone, is what leads to transmission risk of *B. anthracis*, and for brevity we decided to only discuss one set of models.
- Further, while we agree that parametric approaches may be more straightforward to interpret, the purpose behind the GAM was to help identify patterns that parametric models may have missed and to be able to make direct comparisons to the Turner et al. (2014) Etosha study, which this research is attempting to build on.
- Some terminology has been edited to be clearer – “generalized linear mixed models” is now “linear mixed models” or LMMs throughout the manuscript
- We also acknowledge that the multitude of statistical analyses used may be confusing for readers. We have added a summarizing methods table at Line 380 to attempt to make this clearer.

Specific comments

Lines 10 – 11: What is an “extended period of time”? Hours, days, months?

- *B. anthracis* can survive in soil for years and in some cases, decades. A statement describing this has been added in lines 10-11.

Lines 38 – 47: I think this is an extremely interesting question and the authors lay it out really well in the introduction. However, I do think 1 – 2 additional sentences are necessary to clarify an issue the study design. The authors hypothesize (based on a wealth of literature) that animals may avoid LIZs because of cues associated with a carcass in of itself. I have no background in this field, but is it possible that animals could detect the pathogen itself and possibly avoid real LIZs but be attracted to “proxy LIZs”? Obviously, there is no responsible way to test this in the field, but it seems the question should be posed somewhere.

- It is likely difficult for herbivores to directly detect microparasites like bacteria or viruses, which is why herbivores are thought to rely on the detection of parasite-associated cues – i.e. the presence of feces or carcasses. According to this logic,

an individual animal should be similarly avoidant of an infected carcass and a non-infected carcass due to the unknown possibility of pathogen exposure at each. Then, the trade-off an individual makes between grazing at a higher-nutrition carcass site and a lower-nutrition non-carcass site comes down to other factors, like individual characteristics, availability of resources, etc.

- Further, prior work by Turner et al. (2014) found the opposite – that in some cases, herbivores are attracted to anthrax-positive carcass sites.
- We have added to Lines 19-20: “Rather than detect microparasites directly (especially bacteria and viruses), herbivores typically rely on parasite-associated signals.”
- Added to Lines 24-25 (added portion in italics here): Carcass sites hypothetically offer similar cues of possible pathogen presence, so they may be avoided by herbivores, *regardless of the actual infection status of the carcass.*

Methods

Line 143: List the specific variables in parentheses that were tested with the Mann-Whitney U tests.

- The specific variables have been added to Lines 150-151.

Line 152: What was the assumed distribution for the response variable in the generalized linear mixed models and why log transform these variables unless the authors assumed a normal distribution? If they assumed a normal distribution, then the models are not “generalized linear models” – they are just “linear mixed models” or “general linear mixed models”.

- In the methods and results, “GLMMs” has been changed to “linear mixed models (LMMs)” and an additional sentence explaining their use has been added at (what is now) line 160.
- The variables did not have a normal distribution, and thus the log transformation has been removed. This section has been edited to reflect those changes.

Line 165: Why were logistic regressions used to test for correlations? Why not use correlation metrics?

- “Testing for correlations” was the incorrect phrasing to use here. This line has been revised: “Generalized linear models (GLMs) were employed to identify and characterize the effects of a variety of time and season-related predictor variables on the probability of grazing by ungulates.” Now lines 176-178

Line 170: If by Julian Day they mean a numerical day of the year from 1 to 365, then it should be called ordinal day.

- Julian Day has been changed to ordinal date throughout.

Line 181: This could be written like this if they are talking about a logistic model:

$$\text{logit}(p) = \alpha + f_1(x_1) + f_2(x_2) + \dots + f(x_p) + \epsilon_{it}$$

where p represents the probability that...blah blah blah.

- We recognize these formulas were confusing and they have been revised.

Line 191: Is N_t the number of animals that appear in a photo per day?

- Yes, N_t is the total number of herbivores that are present in photos per day – we clarify with revised formulas and Lines 204-207.

Results

The authors tested N, pH, K, P, Cu, Na, and Mg according to the Methods. A quick mention in this section that N, pH, Cu, and Mg were not significant would provide some clarity.

- A sentence clarifying that N, pH, Cu, and Mg were not significant has been added at Line 251.

Lines 225, 229, and 232: Include parameter estimates for Na and pH.

- Parameter estimates have been added here (now lines 260, 264, 266).

Lines 256 – 260: A negative binomial model was used to examine the number of grazing events for each species. I don't remember this in the Methods and I see no other mention of "negative binomial" and "number of grazing events" in the Methods. This is a great example of using consistent terminology for dependent variables between the Methods and Results.

- Lines 256-260 have been moved to be included with (i) Generalized linear models in the results section. More details about this part of the analysis has been added to the Methods section lines 184-188.

Lines 258 – 259: Is this the first mention that the anthrax risk period is June – August?

- For clarity, an additional statement has been added in the methods section, lines 127-130: "Each picture automatically recorded the date and time of capture; this information was used for both diurnal activity assessments and seasonality analyses. The summer months (Jun–Aug) were of particular interest because these months are the historical anthrax outbreak season in the region, dubbed the 'anthrax risk period'".
- The anthrax risk period is also referenced in the results section (lines 299-300) and the description of figure 4 (line 326).

Line 261: I would replace "generalized linear model" with mixed logistic regression where appropriate. It defines the model with fewer words.

- We agree that switching between “logistic regression” and “generalized linear model” was confusing; this method will uniformly be referred to as “generalized linear model” or “GLM” throughout.
- We feel that referring to this GLM as a “mixed logistic regression” would be incorrect here; the model did not include mixed effects, only fixed effects.

Lines 297 – 299: What are the authors talking about here with “host death”?

- We agree that “after host death” was confusing to use here; this has been changed to “as the carcass aged” or “animal death”.

Reviewer: 2

Comments to the Author(s)

Authors are studying 13 localities of a southwestern ranch in Montana, USA to show how minerals and vegetation have higher concentrations on sites with carcasses than areas without. Authors chose this area of study for being a previous focus of an anthrax outbreak despite the lack of current anthrax cases due to control strategies. Thus, they are showing a proxy of anthrax local suitability around non-infected carcasses. The manuscript is well written and evidence is well presented. A couple of comments that should be definitely address:

Although a map is provided. I strongly recommend adding decimal coordinates of the localities studied, as they can be helpful to future spatial-oriented studies.

- This study area is privately owned, and research is managed through a research manager. The coordinates can be provided by reasonable request. As any future research would require land access, such a request would be required. The ranch wildlife biologist, Val Asher (co-author) and the wildlife research office maintain records of research for the private ranch and data are not lost, though the landowner prefers provide information on a request basis.

Line 59-60: Indicate the reference of the landscape characteristics defining the study area, that is, the scrubland, grasslands, coniferous forest, and deciduous forests categorization.

- Landscape characteristics were determined using the 2016 National Land Cover Database; this reference has been added.

Line 181: Please add a space after the symbol of ‘Error term’ in the equation.

- A space has been added.

Line 210: Sometimes you use the word to describe the soil elements and others the symbol (Phosphorus vs P) use either one or the other and be consistent across the manuscript.

- When the soil elements are first introduced (Lines 122-123), both the abbreviation and the full name are included for clarity. For the rest of the manuscript, elements have been changed to abbreviations only (e.g. P, Na, etc.)

Line 211: Why potassium is reported as statistically significant with a $p = 0.08$? If a threshold of significance different than 0.05 was established, this should be stated in the methods section. Otherwise this should be non-significant. Why the value reported here is W instead of U as in the other cases?

- The reviewer is correct; 0.05 was established at the threshold of significance. K was considered marginally significant, at a p -value >0.05 and <0.1 . Lines 239, 247-248 have been edited to more clearly differentiate between the marginal significance of K and the significance of P and Na.
- The use of W was a typo; it has been corrected to U.

Supplementary Figure 1 legend: Consider adding letters to each panel, in the main text you are referring to supplementary figure 1G which lacks the 'G' for identification (line 225; see also line 229 referring to supp. fig. 1F). In line 225, referring to the same figure, notice that the distance of 7.5 m is not statistically significant ($p = 0.06$) but it is not mentioned in the main text.

- A mistake in formatting removed the original letters for each panel; they have been added back.
- Line 259-262 have been modified to state the following: "Model estimates indicate that soil pH increased with distance from the carcass center (1.5 m: std. $\beta = 0.37$; SE = 0.13; $p < 0.01$. 4.5 m: std. $\beta = 0.46$; SE = 0.13; $p < 0.001$. Supplementary Fig. 1G), but the trend did not extend to all sampling zones, as the pH at 7.5 m was not significantly different from that at 4.5 m (7.5 m: $p = 0.06$)."

Anthrax risk period as depicted in Figure five has never been mentioned or referenced (see line 395) in the manuscript. It will be important to reference and comment this specific time interval for interpretation purposes.

- For clarity, an additional statement has been added in the methods section, lines 127-130: "Each picture automatically recorded the date and time of capture; this information was used for both diurnal activity assessments and seasonality analyses. The summer months (Jun–Aug) were of particular interest because these months are the historical anthrax outbreak season in the region, dubbed the 'anthrax risk period'".
- The anthrax risk period is also referenced in the results section (lines 299-300) and the description of figure 4 (line 326).

Figure 3: In the legend please specify that left box plots (or red box plots) correspond to the local infectious zones. Notice that in the first column the x axis for LIZ is named as 0 m and in the second column as 0-3 without unit. Please use the same labels for both columns (e.g., LIZ vs Control) and add any extra detail (0-3 m for biomass quadrat) in the legend.

- Figure 3 has been edited so that axis labels refer to “LIZ” or “Control” only; the explanation of 0 m and 0-3 m has been added to the figure description.
- Additionally, units for each soil element have been added to both Figure 3 and Supplementary Figure 1.

Line 338: Please add a reference to support the exsanguination of anthrax carcasses.

- References have been added.

Line 359: This should be supplementary figure 1. Please revise.

- The reviewer is correct, this has been revised.

Line 373: Supplementary figure 1. Please revise.

- The reviewer is correct, this has been revised.

Line 395: Add a reference for the ‘anthrax risk period’

- References have been added.

I also would like to encourage to add any of the 'cover' pictures as part of the manuscript main text.

Appendix C

Response to Reviewers

Thank you for the reply and acceptance. We have made the minor revisions requested, reformatted the references, and made all of the figure and table changes requested. Here we upload a clean and track changes copy of the manuscript for final editorial changes before publication.

Thank you

Jason K. Blackburn, PhD on behalf of all the authors.